# Effects of the Q80K Polymorphism on the Physicochemical Properties of Hepatitis C Virus Subtype 1a NS3 Protease

**DOI:** 10.3390/v11080691

**Published:** 2019-07-30

**Authors:** Allan Peres-da-Silva, Deborah Antunes, André Luiz Quintanilha Torres, Ernesto Raul Caffarena, Elisabeth Lampe

**Affiliations:** 1Laboratório de Hepatites Virais, Instituto Oswaldo Cruz/FIOCRUZ, Rio de Janeiro, RJ 21040-900, Brazil; 2Programa de Computação Científica, Instituto Oswaldo Cruz/FIOCRUZ, Rio de Janeiro, RJ 21040-900, Brazil; 3Laboratório de Biologia Computacional e Sistemas, Instituto Oswaldo Cruz/FIOCRUZ, Rio de Janeiro, RJ 21040-900, Brazil; 4Laboratório de Bioinformática, Departamento de Bioquímica, Instituto de Química, Universidade Federal do Rio de Janeiro/UFRJ, Rio de Janeiro, RJ 21941-909, Brazil

**Keywords:** hepatitis C virus, Q80K variant, polymorphisms, NS3 protease, simulation, molecular dynamics, network analysis

## Abstract

Hepatitis C virus genotype 1a (HCV-1a) comprises clades I and II. The Q80K polymorphism is found predominantly in clade I but rarely in clade II. Here, we investigated whether natural polymorphisms in HCV-1a clade II entailed structural protein changes when occurrence of the Q80K variant was simulated. Based on HCV-1a clade I and II protein sequences, the structure of the HCV-1a Q80K mutant NS3-4A was obtained by comparative modeling. Its physicochemical properties were studied by molecular dynamics simulations and network analysis. Results demonstrate that, in the presence of the K80 variant, clade II protease polymorphisms A91 and S/G174 led to variations in hydrogen bond occupancies. Structural analyses revealed differences in (i) flexibility of the H57 catalytic residue on the NS3 protease and (ii) correlations between amino acids on the NS3 protease and the NS4A cofactor. The latter indicated possible destabilization of interactions, resulting in increased separation of these proteins. The present findings describe how the relationships between different HCV-1a NS3 protease amino acid residues could affect the appearance of viral variants and the existence of distinct genetic barriers to HCV-1a isolates.

## 1. Introduction

Hepatitis C virus (HCV) represents a global public health threat due to its high rate of evolution to chronic infection and the lack of a vaccine to prevent infections [1]. HCV is classified into seven major genotypes and multiple subtypes [2], of which HCV genotype 1 (HCV-1) is the most prevalent worldwide. Effective therapeutic drugs directly targeting HCV proteins, the so-called direct-acting antivirals (DAAs), have been the subject of intense research in the last decade. As the HCV NS3-4A protease is a critical component of the viral replicase complex, several protease inhibitors have been developed for use in clinical practice [3,4,5,6].

HCV displays high genetic heterogeneity and great sequence variability, contributing to the existence of natural polymorphisms. HCV is present within patients as a *quasispecies*—variants that are closely related but heterogeneous [7]. Variability of the HCV genome poses a challenge for DAAs, because drug resistance represents the leading cause for the failure of antiviral therapy against HCV infections. The Q80K polymorphism is found predominantly in HCV genotype 1a (HCV-1a) sequences and has been associated with reduced inhibitor activity of not only simeprevir [8], but also other protease inhibitors in both in vitro and in vivo experiments [5,9]. Additionally, this polymorphism could facilitate the emergence of resistance-associated substitutions (RAS) conferring the virus increased resistance to protease inhibitors or other DAAs [9,10].

Based on 282 statistically significant phylogenetic informative sites in the HCV genome, HCV-1a has been differentiated into two clades, I and II [11]. Several sites responsible for this distinction are located near or within codons associated with resistance to protease inhibitors, which would increase the possibility of one of these HCV-1a clades being more likely to develop resistance to DAA inhibitors. This same pattern of segregation was observed in HCV NS3 protease domain sequences containing 19 phylogenetic HCV-1a clade informative sites, whereby the Q80K polymorphism was detected in most HCV-1a clade I sequences but was rarely present in clade II [12]. The two clades have a different geographic distribution, with clade I accounting for 75% of the US sequences but less than half of the European ones. Consequently, the prevalence of the Q80K polymorphism is higher in the US than in Europe [13]. The vast majority of Brazilian HCV-1a sequences are grouped in clade I, mostly as a separate group of related sequences with a high supporting value (subclade IC). However, the prevalence of the K80 polymorphism in this population is very low (≤3%) within this subclade [12,14]. Accordingly, the presence of polymorphic sites in HCV genotypes, subtypes, and even in HCV-1a clades may be critical for the different Q80K frequencies observed in HCV NS3 sequences.

This study applied molecular dynamics (MD) simulations and network analysis to investigate whether the distinct natural polymorphisms in HCV-1a sequences caused structural changes in HCV-1a clade II proteins when the occurrence of the Q80K variant was simulated. Results point to the possible adverse effect of combining the Q80K variant and naturally occurring polymorphisms in clade II on the structural and enzymatic properties of the NS3 protease. Thus, polymorphisms could pose genetic barriers to the appearance of viral HCV-1a isolates, explaining why Q80K variants are not detected in HCV-1a clade II sequences.

## 2. Materials and Methods

### 2.1. Sequence Analysis and Statistical Treatment

A total of 120 HCV-1a clades I and II reference sequences per clade were extracted from Pickett et al. [11]. The sequence from the entire NS3 protease clade I domain (181 amino acids) was screened and compared to that of NS3 protease clade II using T-coffee [15]. Sequences were aligned with the Vespa program [16] to determine the subtype-clade-specific amino acid signature pattern at NS3 protease sites.

Based on the HCV-1a clade I and II amino acid signature pattern, only clade I sequences presented high frequencies of K80, S91, and N/S174. To verify if K80+S91+N/S174 polymorphisms were more frequent in the same clade I sequences than based on a random distribution, we estimated the frequencies of K80+S91+N174 and K80+S91+S174 and then compared the results with the frequency of clade I Q80+S91+N174 and Q80+S91+S174 polymorphisms. To this end, we considered 116 sequences, which contained either K or Q in position 80; the remaining four sequences had either G, L, N, or R at this site. A chi-square test was performed to verify the significance of sample size, with *p* < 0.05 being considered statistically significant. The total sample size of Q80 (44/116, 37.9%) and K80 (72/116, 62.1%) polymorphisms and their frequencies was estimated. A chi-square test was used to compare the observed counts of each category (K80+S91+N174, Q80+S91+N174 and K80+S91+S174, Q80+S91+S174) with their respective expected counts, specified by probability distributions.

Four mutant NS3-4A sequences (two for each clade) were retrieved from GenBank for subsequent comparative modeling. The frequencies of clade I K80, S91, and S/N174 polymorphisms resulted in the inclusion of GenBank sequences EF407438 (c1mutKSN = clade I, K80+S91+N174) and EU155215 (c1mutKSS = clade I, K80+S91+S174). Clade II sequences presented high frequencies of A91 and G/S174, resulting in selection of GenBank sequences EU256049 (c2mutKAG = clade II, K80+A91+G174) and EU239713 (c2mutKAS = clade II, K80+A91+S174). Importantly, these HCV sequences showed variations only at the 91 and 174 clade-polymorphic sites. Sequences presenting additional polymorphisms at other sites could interfere in the analysis of the physicochemical properties of mutant proteins and hence were excluded.

As K80 was detected only in NS3 sequences belonging to HCV-1a clade I, we considered such sequences and the ensuing mutant structures’ dynamic behavior as reference controls. For HCV-1a clade II sequences, a Q80K alteration was made in clade II GenBank EU256049 and EU239713 sequences prior to molecular modeling. A comparison with HCV-1a clade I controls, determined if this introduction of K80 affected HCV NS3 physicochemical properties.

### 2.2. Comparative Modeling

The BLASTP program was used to search the Protein Data Bank (PDB) [17] and select a template structure to be used in the comparative modeling procedure. A single template, 2f9v [18], was used for all target sequences generated. This specific template was chosen based on identities and coverage between target and template sequences. PDB crystallographic resolution (2.6 Å) and co-crystallization of the NS4A cofactor with the template structure were also taken into account.

The template and target sequences were aligned using the PSI-Coffee mode in T-Coffee. A hundred homology models were generated for each target sequence using the standard “auto model” routine in Modeller version 9.18 [19]. Each model was optimized using the variable target function method until accomplishing 300 iterations. MD optimization was carried out in the slow level mode. The full cycle was repeated twice to generate an optimized conformation of the model. The best resulting modeled structures were selected according to their discrete optimized protein energy (DOPE) score. The models’ quality was evaluated using DOPE, ERRAT2 [20], Ramachandran plots, and the QMEAN server [21]. Sequence alignments were rendered using ALINE [22]. Three-dimensional (3D) structure figures were generated using the UCSF Chimera program [23].

### 2.3. Molecular Dynamics (MD) Simulations

MD simulations were carried out using the GROMACS 5.1.2 package [24], and protein interactions were represented using the AMBER99SB ILDN [25] force field. A set of 100-ns production runs was performed to thoroughly investigate the dynamic behavior of HCV-1a NS3-4A protease mutant proteins. Protonation states were assigned using pdb2pqr software, and zinc-coordinating cysteine residues were manually deprotonated to maintain ion coordination. Electrostatic interactions were treated using the particle mesh Ewald algorithm with a cut-off of 10 Å. Each system was simulated under periodic boundary conditions in a triclinic box, whose dimensions were defined automatically based on the most distant atoms from the center of the protein’s mass being 1 nm away from the box edges and the box being filled with TIP3P water molecules [26]. All systems were neutralized by adding counterions.

Simulations were performed in three stages: (i) energy minimization, (ii) thermalization and equilibration, and (iii) trajectory production.

Energy minimization was achieved through 3000 steps and a gradient tolerance <1.0 kJ mol^−1^ of the steepest descent (with and without heavy atom restraints by applying a harmonic potential with a force constant of 1000 kJ mol^−1^ nm^−2^) and conjugate-gradient algorithms.

In the second phase, starting atomic velocities were assigned to all atoms in the system using a Maxwell-Boltzmann distribution, corresponding to an initial temperature of 20 K. The temperature was gradually increased to 300 K over 500 picoseconds (ps) utilizing the Langevin thermostat. During this stage, all heavy atoms were restrained by applying a harmonic potential with a force constant of 1000 kJ mol^−1^ nm^−2^.

Systems were subsequently equilibrated during four successive equilibration simulations of 750 ps each, corresponding to different force constant values (900, 750, 500, 250 kJ mol^−1^ nm^−2^) from harmonic restraints. After this period, simulations ran with no restraints for 100 ns. All simulations were performed in the NPT ensemble. The V-rescale thermostat and Berendsen barostat were used for temperature (300 K) and pressure (1 atm) control, respectively. We performed five replicas simulations on the protease systems, each of 100 ns, starting from the same structure and running the simulation independently, yielding a total simulation time of 500 ns.

### 2.4. Trajectory Analysis

Differences in structural stability and dynamic nature of the mutants were investigated by comparing structural protein descriptors. Root-mean-square deviation (RMSD) and root-mean-square fluctuation (RMSF) values were calculated after taking the initial structure from production dynamics as a reference. VMD software was used to calculate distances between atoms and determine hydrogen bond (H-bond) formation using a geometrical criterion. A hit was computed when the distance between the heavy atom acceptor and the hydrogen atom was <2.5 Å and an angle <30° was formed between the acceptor, hydrogen, and donor atoms.

### 2.5. Correlation Network Analysis

Cross-correlation and network analyses were carried out in R [27] with the Bio3D program and the igraph package [28]. Initially, dynamic cross-correlation matrices (DCCM) were calculated separately for each simulation using as input the corresponding last 70 ns of the MD trajectory superimposed onto the initial structure. DCCM was used for building the residues correlation network. Briefly, graphs were obtained considering Cα atoms as nodes, and the connection between nodes i and j was weighted by the absolute values of cross-correlation (*C*_(i,j)_) coefficients:*w*_(i,j)_ = −*log*(|*C*_(i,j)_|).

The community analyses network was built using the Girvan-Newman algorithm [29].

### 2.6. Phylogenetic Analysis

To investigate possible correlations between clade I S91+N/S174 polymorphisms and the K80 phenotype, phylogenetic analysis was performed with the same HCV-1a clade I and II reference sequences used to determine specific amino acid signature patterns. Maximum-likelihood phylogenetic trees were inferred using the PhyML program [30] with the approximate likelihood-ratio test (aLRT) [31] based on a Shimodaira–Hasegawa-like procedure. Specifically, general time reversible was used as substitution model and a subtree pruning and regrafting-based tree search algorithm was used to estimate tree topologies.

## 3. Results

### 3.1. Sequence Analysis, Statistical Treatment, and Comparative Modeling

Different amino acid signature frequencies were documented at positions 80, 91, and 174 of NS3 protease across HCV-1a clades. Q80 (44/120, 36.7%), K80 (72/120, 60.0%), S91 (43/120, 35.8%), N174 (97/120, 80.8%), and S174 (22/120, 18.3%) were found at high frequencies in HCV-1a clade I. In clade II, the most prevalent residues at these sites were: Q80 (117/120, 97.5%), A91 (116/120, 96.7%), S174 (86/120, 71.7%), and G174 (15/120, 12.5%). These polymorphic frequencies are in accordance with a recent analysis of more than 2000 HCV-1a NS3 sequences from both clades [13].

We observed 30 isolates containing S91+N174, of which 12 (40.0%) were K80+S91+N174 and 18 (60.0%) were Q80+S91+N174 (*p* = 0.0127). Another 13 isolates contained S91+S174, of which, 12 (92.3%) were K80+S91+S174 and one (7.7%) was Q80+S91+S174 (*p* = 0.0246). It should be noted that there were also sequences containing K80 (*n* = 48), Q80 (*n* = 25), G80 (*n* = 1), L80 (*n* = 1), N80 (*n* = 1), and R80 (*n* = 1) residues. These sequences presented a broad polymorphic variety in their NS3 sequence, including at sites 91 and 174, and were not incorporated in the structural analysis of the present study.

To understand the structural implications of these polymorphisms in conjunction with K80, we built 3D models of four HCV-1a clade proteins, whose amino acid composition varied only at sites 91 and 174. As no Q80K variation was identified in HCV-1a clade II sequences deposited in GenBank, we manually altered Q80K in the 2 HCV-1a clade II sequences (c2mutKAS and c2mutKAG) prior to molecular modeling.

From the set of 57 structures returned by the BLASTP program, we chose the HCV NS3 protease domain with NS4A peptide (PDB ID: 2f9v) [18] as template. The template sequence shared 97% identity with clade II sequences and 98% identity with clade I sequences, based on 100% sequence coverage (Figure 1). An evaluation of HCV NS3-4A protease models is presented in Table 1, and a 3D representation is shown in Figure 2.

### 3.2. Assessing Dynamic Properties of the Models in Aqueous Solution

An important aspect of MD simulation analysis is the description of macromolecular motions. The most common mobility measures are the RMSD and RMSF, obtained after aligning the atomic coordinates in each trajectory step to a reference structure. The RMSD is used for the evaluation of differences between molecules. It can be used to calculate the global deviation during the trajectory and evaluate the oscillation of the system during the elapsed time. Uniform RMSD values that remain below a certain limit (this limit depends on the protein) indicate that the structure deviates only marginally from the initial position, whereas abrupt changes in mean values denote important changes in structural conformation.

The RMSF, a measure complementary to the RMSD, allows to obtain information about local structural flexibility. The RMSF calculates the atomic positions throughout the simulation. Its values might reflect only displacements of a small structural subset within an overall rigid structure.

According to our findings, during MD simulations, the RMSD values of all HCV mutants’ NS3 protease maintained relatively consistent deviations of approximately 1.0–1.5 Å and achieved stabilization after ~35 ns (Figure 3a). For the NS4A cofactor, all systems except c2mutKAG achieved high RMSD values (>3.0 Å) after ~35 ns (Figure 3b).

To investigate the effect of the different mutations on residues’ dynamic behavior and stability, the RMSF of each HCV-1a mutant protease residue was calculated (Figure 4). HCV NS3 protease residues 80, 91, and 174 did not present substantial fluctuations around their respective averages of 0.8 Å, 1.19 Å, and 0.58 Å. However, residues 14 and 15, which are part of an alpha helix at the N-terminus of the protease, showed fluctuations >2 Å in the c1mutKSS structure. These residues are located close to the NS4A cofactor, which is probably the reason for the elevated fluctuation values observed for this structure. Higher fluctuations were observed also for residues 89 (2.38 Å) and 90 (2.22 Å) of the c2mutKAS structure, as well as residue 161 of c1mutKSN (2.61 Å) and c1mutKSS (2.33 Å), and residue 181 (3.36 Å) of c2mutKAG. Regarding protease catalytic residues, c2mutKAS and c2mutKAG structures exhibited more flexibility for H57, whereas the c1mutKSN structure displayed high flexibility for S139. These differences in flexibility likely allowed accommodation of the surrounding polymorphisms at sites 80, 91 or 174 and the catalytic residues in these mutated structures. No flexibility differences were observed for the catalytic residue D81 in any of the systems. In the NS4A cofactor protein, residue 182 and the final residues (198–203) of the c2mutKAG structure exhibited fluctuations >3 Å.

### 3.3. Hydrogen Bonding Interactions

To understand the possible effect of the Q80K mutation on HCV-1a clade II proteins’ interactions, intramolecular H-bonds were calculated for all systems and compared to those of HCV-1a clade I control structures. H-bond occupancy between atoms from residues involved in the interaction is summarized in Table 2. The presence of the resistant amino acid K80 in the c2mutKAS structure resulted in the formation of nine H-bond interactions between side-chain atoms of catalytic residues H57 and D81 (~96%). Among these H-bonds, H^57^ (ND1) − D^81^ (CG), H^57^ (ND1) − D^81^ (OD1), and D^57^ (ND1) − D^81^ (OD2) presented high occupancy values of 48.99%, 27.41%, and 20.04%, respectively. The same H-bonds were observed in the control c1mutKSS structure, but at much lower occupancies of 10.34%, 6.38%, and 7.64%, respectively. The RMSF of residue 57 in the c2mutKAS structure was also higher (1.00 Å) compared to the control c1mutKSS (0.71 Å). The higher RMSF may have allowed a greater proximity to residue 81, as evidenced by the shorter distance between atoms involved in H-bond interactions (Table 3), and consequently resulted in higher H-bond occupancies.

Two H-bond interactions between atoms were formed between the catalytic side-chain atoms of S139 and those of backbone K136. In the c2mutKAG case, the occupancy in one of these interactions, S^139^ (OG) − K^136^ (O), was substantially lower (6.58%) than in the other structures (Table 2). RMSFs of residues located at positions 136 (0.89 Å) and 139 (0.49 Å) were also slightly lower in comparison to the control c1mutKSN, c1mutKSS, and c2mutKAS structures, whose respective RMSF values were as follows: 1.21 Å, 1.00 Å, and 1.02 Å (residue 136) and 1.04 Å, 0.52 Å, and 0.54 Å (residue 139). Hence, the slight reduction in flexibility affecting residues 136 and 139 in c2mutKAG may have prevented sufficient closeness and H-bond formation between these two residues. The greater distance between oxygen atoms in these two residues (5.2 Å) (Table 3) and the lower H-bond occupancy appeared to confirm this hypothesis.

Differences observed for H-bond interactions involving the catalytic triad residues of HCV-1a clade II mutated structures could affect NS3 protease activity, providing evidence that changes in chemical bonding patterns in these structures would be a plausible reason for the non-detection of Q80K variants in HCV-1a clade II sequences.

### 3.4. Correlation Network Analysis and Community Network Comparison

A strategy based on network analysis was applied to study motion correlations between residues and among the four mutated HCV-1a clade structures. In this approach, each residue represented a node and the weight of the connection between nodes, i and j, represented their respective cross-correlation value, *w*_ij_. We analyzed how the simulated occurrence of the Q80K mutation in HCV-1a clade II sequences affected correlations between residues in NS3-4A protease structures and compared them to correlations between HCV-1a clade I mutants.

Overall, a similar pattern of cross-correlations was observed among all systems (Figure 5). Both HCV-1a clade II mutant structures displayed anticorrelations between residues belonging to the NS4A cofactor and the NS3 protease. Anticorrelations were more pronounced in the c2mutKAG structure, indicating that the affected residues moved in opposite directions. These results were consistent among replicas, indicating that the observed differences were real and reproducible. Notably, the only difference between the HCV-1a clade II structures used in this study were the residues located at site 174, which are close to the NS4A cofactor. The presence of G174 was likely responsible for the increased anti-correlations in the c2mutKAG structure. DCCM confirmed that the movement between the HCV NS3 protease and the NS4A cofactor was opposite to what was observed for HCV-1a clade I K80 control structures.

Community network analysis revealed significant differences between HCV clade I and clade II mutated proteins (Figure 6). Correlations between the HCV NS3 protease and the terminal residues (197–203) from the NS4A cofactor were clearly evidenced in the simulated clade II systems. In the case of c2mutKAS, these terminal residues formed a community of 39 residues (4:7, 46, 53:62, 73, 75:83, 136, 191:203) that included the protease K80 resistance variant residue, as well as catalytic residues 57 and 81. In the c2mutKAG system, the 197–203 cofactor residues were included in a community of 67 residues (4:9, 24:45, 63:72, 92, 108:112, 138, 182:203). These terminal cofactor residues formed a single and separated community in the c1mutKSN and c1mutKSS systems. Therefore, simulation of the K80 mutation in clade II systems demonstrated the generation of a new correlation pattern with the NS4A cofactor, which may not guarantee proper enzymatic functioning in these mutated structures.

To quantify the relative importance of each residue in the network, we computed the betweenness centrality per amino acid for each simulated system (Figure 7). Betweenness centrality is a measure based on the number of shortest paths between any two nodes. A high betweenness centrality might suggest that an individual node is connecting various parts of the network. The catalytic residue H57 displayed higher betweenness in c2mutKAS (0.06) than in the other mutant structures studied. To compare the overall similarity of betweenness centrality profiles, we calculated the square inner product (SIP) (Table 4). A higher SIP was observed when comparing HCV-1a clade I structures (0.62) than HCV-1a clade II structures (0.46).

### 3.5. Phylogenetic Analysis

As shown in Figure 8, the phylogenetic tree was divided into two major clades, with K80+S91+N/S174 sequences clustered exclusively in clade I within two distinct monophyletic subclades, IA and IB. Subclade IA included 16 K80 isolates, half of which contained S91+N174 residues and half S91+S174 residues. This subclade contained also 16 wild-type sequences (Q80+S91+N/S174), which were grouped with high statistical support (aLRT = 0.90). In contrast, subclade IB comprised 27 HCV-1a sequences, of which four presented the polymorphic sequence K80+S91+N174 and another four the K80+S91+S174 sequence.

## 4. Discussion

HCV-1a comprises two clades (I and II), with the Q80K polymorphism being found predominantly in clade I and only rarely in clade II. However, the association between HCV-1a clades and Q80K polymorphism seems to be more complex. As shown by Santos et al. [14], clade I can be further split into at least three subclades (IA to IC). Notably, the Q80K polymorphism is associated with subclade IA and is mostly absent from subclades IB and IC. Most of the HCV-1a sequences reported in Brazil belong to subclade IC and some Argentinean HCV-1a sequences are grouped within clade I. Only a very low prevalence of Q80K has been reported in these countries [12,32]. Not surprisingly, the A91 polymorphism is also predominant in subclades IB and IC, as well as in clade II [14]. This fact suggests that the presence of alanine at site 91 may be related to the absence of lysine at site 80 of these sequences.

In our analysis, Q80K polymorphism correlated with S91+N/S174 substitutions. Importantly, these residues arose in viral lineages of two independent subclades (IA and IB), rather than branching out from a unique ancestral node. Coupling the structural results of the present study with analysis of coevolutionary dependencies between residues corroborates the findings and helps elucidate the relationships driven by interaction correlations.

After 100 ns of MD simulations, RMSD values from HCV-1a clade models of NS3 proteins containing Q80K polymorphisms exhibited only minimal differences among them. However, differences and fluctuations were higher for the NS4A cofactor. This may be explained by the greater exposure of this protein to the solvent. Furthermore, we investigated the effect of the Q80K mutation on the residues of each system. Residues of the catalytic triad, H57, D81, and S139, showed distinct flexibilities when clade I and II mutant systems were compared.

The simulation of Q80K occurrence in the HCV NS3 protease also altered H-bond interactions of HCV-1a clade II proteins. When compared to clade I K80 control systems, c2mutKAS displayed more H-bonds between the catalytic residues H57 and D81. In contrast, in the c2mutKAG system, there were fewer H-bond interactions between the side chain of catalytic residue S139 and K136 on the backbone. Importantly, residues H57 and S139 are structural neighbors of polymorphic sites 91 and 174. These data confirm the possible impact of polymorphisms in HCV-1a clade II sequences on the catalytic triad residues in the presence of the Q80K mutation. Indeed, by affecting the chemical bonds in the catalytic site, the polymorphisms could compromise the protein’s enzymatic activity.

Interestingly, noticeable anticorrelations were observed between the terminal residues of the NS4A cofactor and the NS3 protease in HCV-1a clade II systems, especially in c2mutKAG. Increased anticorrelations may indicate a possible destabilization of the interactions between these proteins, which could ultimately lead to an increased separation between them.

The betweenness centrality metric allowed the identification of critical nodes for network-wide communication. Residues presenting high betweenness values are considered “bottlenecks” of information as they are mostly found in the shortest paths of communication [33]. Interestingly, the catalytic residue H57 displayed a high betweenness value in c2mutKAS, indicating relevant communication through this particular residue. Elevated SIP of betweenness centrality was associated with weak modulation of intramolecular communication, thus reinforcing noticeable Q80K-related effects in c2mutKAG and c2mutKAS systems.

The HCV NS3 A91S and S174N polymorphisms were strongly associated with Q80K in a previous study by McCloskey et al. [34]. However, Santos and colleagues [14] only found an association between S174N and Q80K in subclade IA but not in subclades IB and IC, despite the high frequency of the S174N polymorphism in these last two subclades. Interestingly, most subclade IB and IC sequences have the same A91 polymorphic pattern as seen for clade II sequences. Santos et al. [14] also proposed that mutations at HCV NS3 80, 91, and 174 sites were not necessarily biologically relevant. However, our results suggest that the presence of the Q80K polymorphism is associated with S91 and N/S174 polymorphisms, which would affect NS3 protease structure and contribute to K80 stabilization. In this sense, natural polymorphisms found in the HCV genome are crucial determinants of viral fitness, and the probability of the Q80K polymorphism in the viral quasispecies population relates to the fitness of the corresponding NS3 protease structures [35].

The NS3 proteolytic activity and consequently viral replication could be compromised by modifying the physicochemical properties of the NS3 protease in the presence of the Q80K mutation, potentially hindering its emergence in the clade II viral quasispecies population and explaining why this mutation is not detected in these sequences.

In vitro studies using HCV subgenomic replicons, infectious cell culture systems and HCV-1a clade I mutants as controls, could elucidate whether the physicochemical differences highlighted in this study led to a decrease in viral replication capacity and reveal the pattern of viral variants that emerges from the different HCV-1a isolates. At present, we did not measure in vitro biological activity and thus could not confirm whether the Q80K mutation in clade II proteins did indeed diminish enzymatic activity.

Different HCV genotype 1a baseline polymorphisms determine the emergence pattern of additional resistance-conferring polymorphisms. While simeprevir treatment favored the emergence of resistant mutations such as D168A/V in the wild-type virus, engineered Q80K/R recombinants were responsible for positively selecting resistant R155K variants [36]. Also, viral fitness depended on these specific substitutions. The replication capacity of Q80K variants was 1.26-fold higher than that of the wild-type virus. However, after the emergence of R155K variants (Q80K+R155K), the replication capacity was only 0.63-fold higher than that of the wild-type virus [36]. These results confirm that baseline polymorphisms present in the HCV genotype 1a NS3 protease are determinants not only of RAS selection during DAA therapies but also of viral fitness. In another study, Pham and colleagues investigated the effects of engineered substitutions at position 80 on the fitness of three HCV genotype 1a infectious recombinants (isolates H77, TN, and HCV1) by comparing viral spreading kinetics. Interestingly, for isolate TN, which originally harbored K at position 80, viral spreading kinetics was significantly delayed compared to that of the original viruses. This isolate (GenBank JX993348.1) was the only one to contain A91 in its sequence [9]. This fact, not addressed in that study, demonstrates that polymorphisms present in HCV genotype 1a sequences differentially impact resistance in viruses. However, no study so far engineered HCV genotype 1a K80+A91+G/S174 recombinants to evaluate viral fitness and the ability to promote additional RAS emergence.

Given the high genetic variability of HCV, other polymorphic sites present in non-structural genes (NS5A or NS5B) could also determine the emergence of DAA-resistant viral variants. Indeed, in vitro studies have demonstrated that baseline polymorphisms can favor the selection of variants resistant to DAA inhibitors. Sun and colleagues used hybrid replicon cell lines to demonstrate that the HCV NS5A polymorphism E62D, which has no effect on daclatasvir inhibitor activity, could not only influence the emergence of the resistant variant Q30R, but was also able to increase the levels of resistance to daclatasvir [37].

As the sequences analyzed here were obtained by Sanger sequencing, we were not able to confirm whether all PCR-amplified molecules from a particular viral isolate contained the triple mutations addressed in this study. Further, the eventual detection of Q80K variants present at low frequencies in HCV-1a clade II isolates may have been missed with this approach. However, it does not necessarily mean that a minor Q80K variant population would prevail over time or that these variants would emerge to become the main viral population without the occurrence of additional mutations in these HCV genomes.

Here, we demonstrate that HCV isolates of the same subtype (1a) exhibit different molecular behavior in the presence of DAA-resistant substitutions. It is quite plausible that other HCV proteins also exhibit a differential molecular behavior pattern. It is not yet fully understood whether the differences in frequencies in DAA resistance substitutions found in HCV non-structural proteins from different geographical regions were caused by changes in physicochemical properties. In this sense, the protocol described here could help predict whether interactions between residues cause changes in HCV protein structures.

The physicochemical differences to the NS3 protease described in this study may be altered by the appearance of additional mutations driven by the elevated replication rates of HCV. In fact, such additional mutations may even cause the physicochemical properties of these proteins to become similar to those of the control structures used in this study.

In conclusion, structure prediction and MD simulations yielded important structural information on HCV-1a clade DAA-resistant proteases. These findings will pave the way for an improved understanding of the atomic interactions between residues and the design of new antiviral therapeutics that can tackle currently resistant variants. Moreover, the methods used in this study are also applicable to other non-structural HCV proteins, whose mutated structures and behavior in an aqueous medium have not been investigated yet by MD simulations.

## Figures and Tables

**Figure 1 viruses-11-00691-f001:**
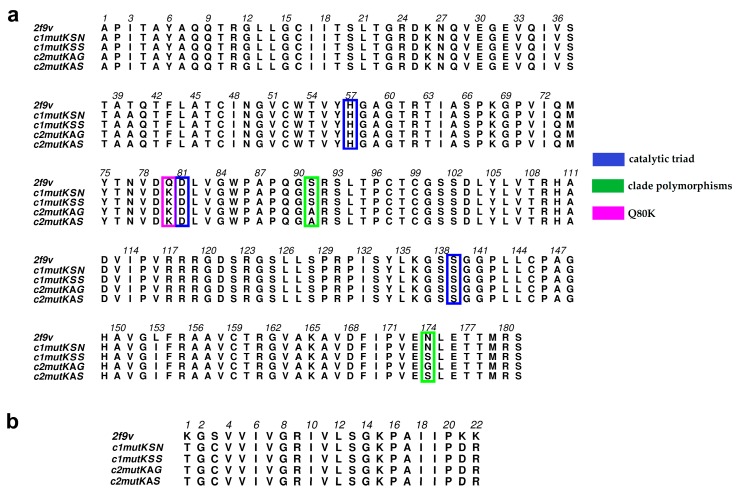
(**a**) Multiple hepatitis C virus (HCV) NS3 sequence alignment between the four hepatitis C virus genotype 1a (HCV-1a) clade protein sequences and the Protein Data Bank (PDB) 2f9v template; positions 80 (pink), 174 (green), 91 (green), and the catalytic residues 57, 81, and 139 (blue) are highlighted. (**b**) Multiple HCV NS4A cofactor sequence alignment between the four HCV-1a clade protein sequences and the PDB 2f9v template.

**Figure 2 viruses-11-00691-f002:**
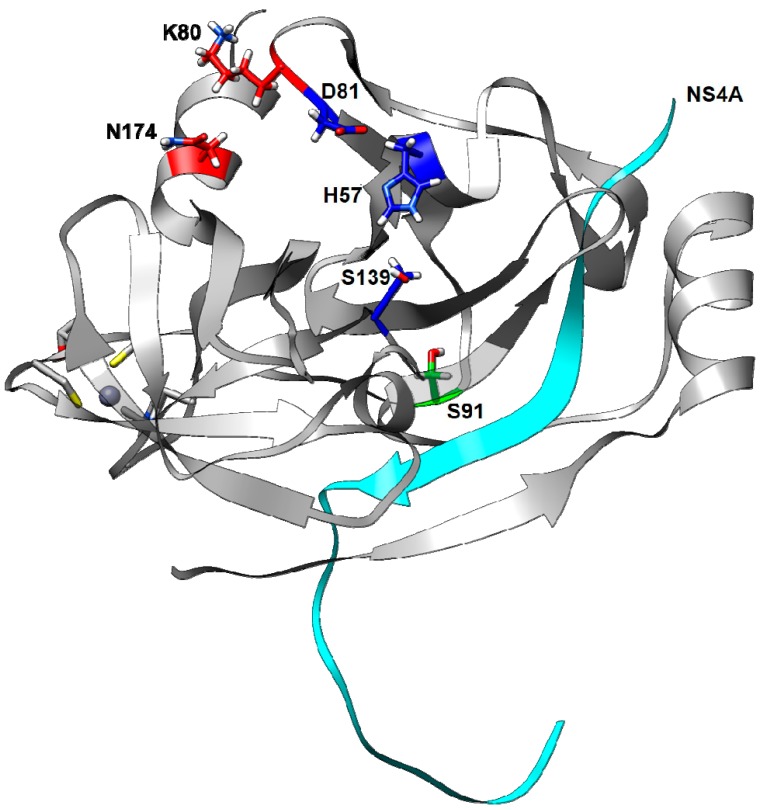
Three-dimensional structure of HCV NS3 protease based on the c1mutKSN sequence: positions 80 (red), 174 (red), 91 (green); the catalytic residues 57, 81, and 139 (blue); and the cofactor NS4A (cyan) are highlighted. Residue 80 is proximal to residue 174, but distant from residue 91.

**Figure 3 viruses-11-00691-f003:**
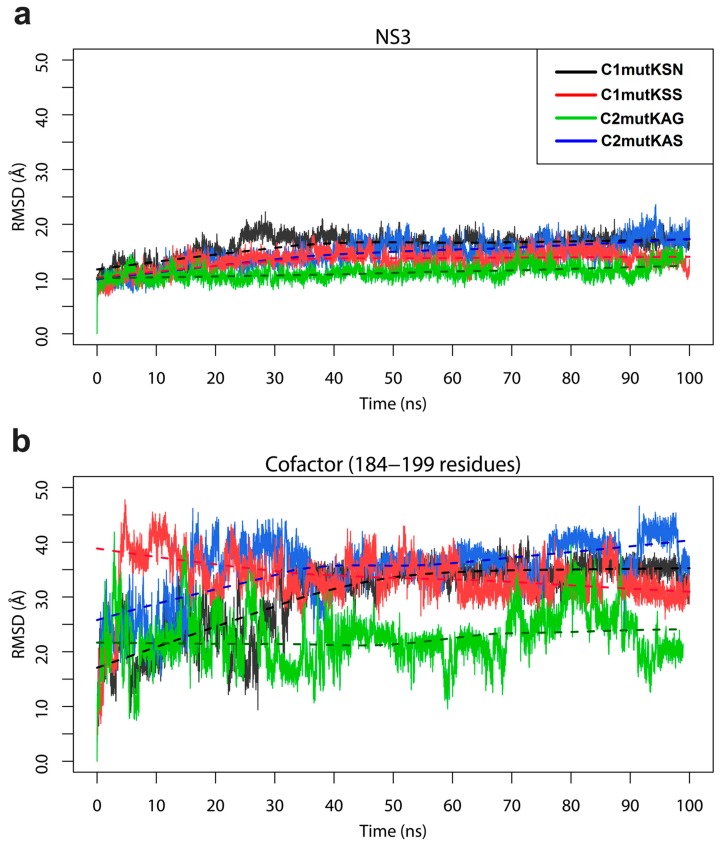
Root-mean-square deviation (RMSD) values of HCV-1a NS3 protease (**a**) and NS4A cofactor (**b**) clade proteins based on molecular dynamics (MD) trajectories.

**Figure 4 viruses-11-00691-f004:**
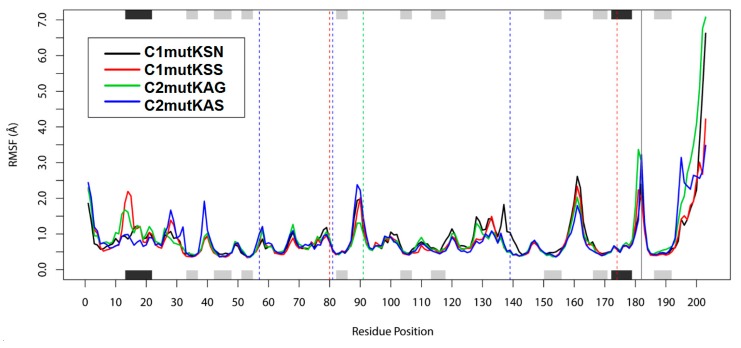
Root-mean-square fluctuation (RMSF) values. The positions of helices (black) and strands (grey) are indicated on the top and bottom axes of the fluctuation plot. Dashed vertical lines indicate residues 80 and 174 (red), 91 (green), and catalytic residues 57, 81, and 139 (blue).

**Figure 5 viruses-11-00691-f005:**
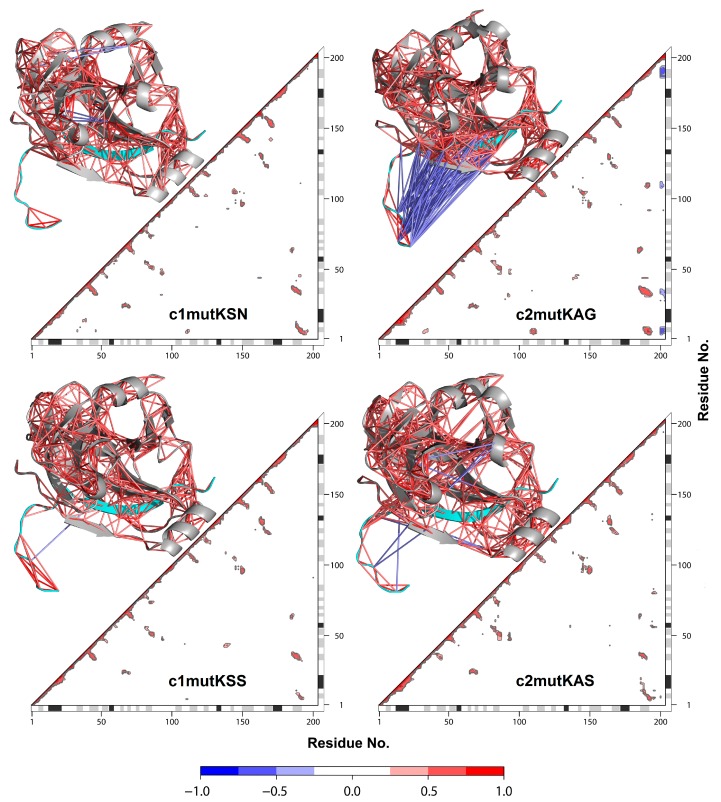
Dynamic cross-correlation matrix with the correlation coefficients module (|*C*_(ij)_|) > 0.4. For each matrix, the 3D structure is shown. Each value ranges from +1.0 to −1.0, representing a pair of amino acids being either completely correlated (red) or anticorrelated (blue), respectively. NS4A cofactor (cyan).

**Figure 6 viruses-11-00691-f006:**
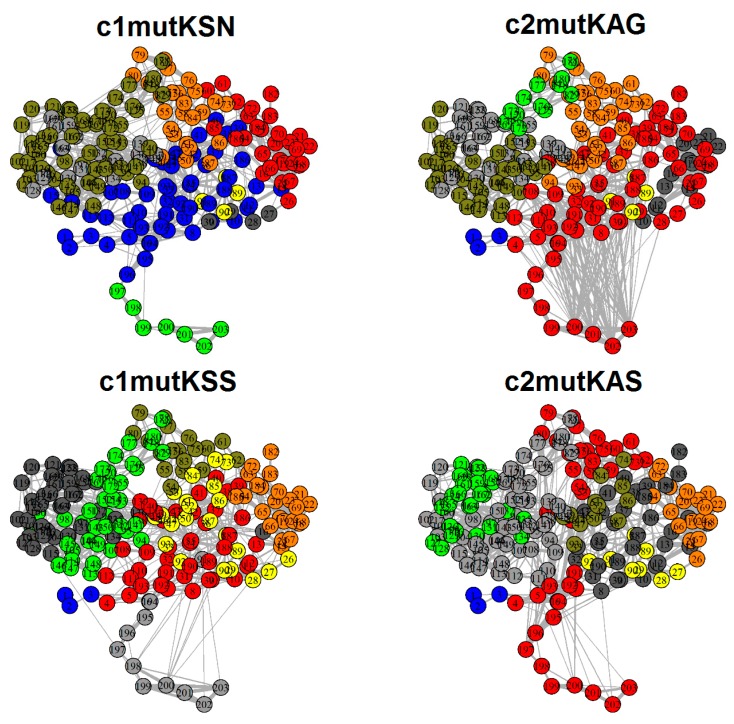
Community networks of the four HCV-1a clade systems. Correlation between NS3 residues and the terminal NS4A cofactor residues (197–203) is evidenced in the simulated clade II systems (red). These terminal cofactor residues form a single and separated community in the c1mutKSN (green) and c1mutKSS (grey) systems.

**Figure 7 viruses-11-00691-f007:**
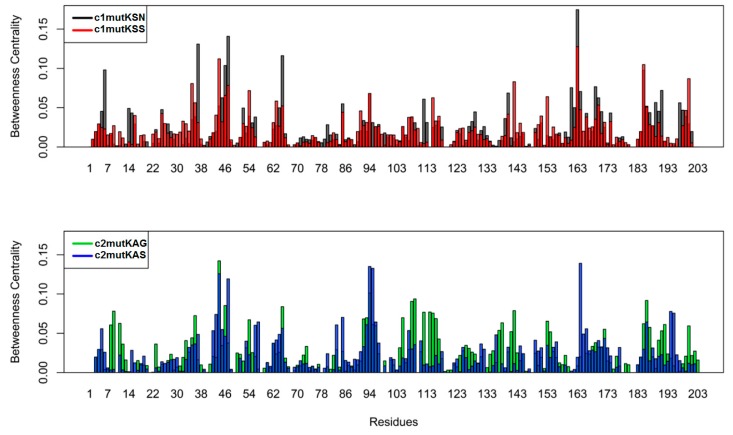
Betweenness centrality of the node for each residue.

**Figure 8 viruses-11-00691-f008:**
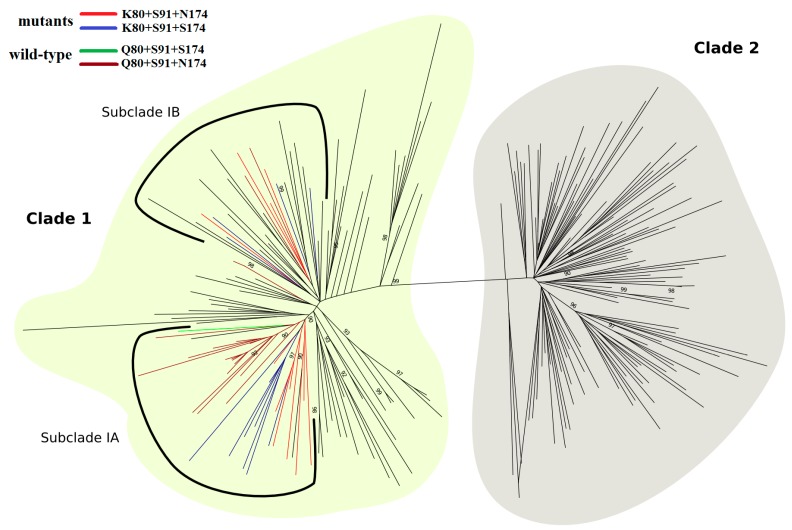
ML phylogenetic tree of HCV-1a based on 120 representative sequences of each, clade I (light yellow) and clade II (gray). Within clade I, subclade IA and subclade IB are highlighted by a solid black line. A total of 12 K80+S91+N174 (red) and 12 K80+S91+S174 (blue) polymorphic sequences are grouped in the two subclades, whereas most of the 19 wild-type sequences (brown and green) cluster into a single branch, formed by 16 sequences. In clade I, branches colored in black are represented by sequences containing K80 (*n* = 48), Q80 (*n* = 25), G80 (*n* = 1), L80 (*n* = 1), N80 (*n* = 1), and R80 (*n* = 1). In clade II, black branches represent sequences containing Q80 (*n* = 117), R80 (*n* = 2), and L80 (*n* = 1). The tree was rooted with HCV-1b strain (D90208). Approximate likelihood-ratio test values ≥0.90 are shown. For better visualization, sequence names were removed.

**Table 1 viruses-11-00691-t001:** Evaluation of HCV NS3 protein models.

Models	DOPE Score	ERRAT2	QMean	Ramachandran Plot		
				R1	R2	R3
2f9v	−22100	94.047	−0.35	95.5	4.5	0.0
c1mutKSN	−21543	83.132	0.19	97.9	2.1	0.0
c1mutKSS	−21453	92.261	0.21	98.5	1.5	0.0
c2mutKAG	−21581	91.071	0.13	98.5	1.5	0.0
c2mutKAS	−21665	89.881	0.44	98.5	1.5	0.0

2f9v, template structure; R1, percentage of residues in favored region; R2, percentage of residues in allowed region; R3, percentage of residues in outlier region; K, lysine; N, asparagine; G, glycine; S, serine; c1mutKSN = clade I, K80+S91+N174; c1mutKSS = clade I, K80+S91+S174; c2mutKAG = clade II, K80+A91+G174; c2mutKAS = clade II, K80+A91+S174.

**Table 2 viruses-11-00691-t002:** H-bond pairs and their occupancy in the HCV-1a clade systems.

Atoms		Occupancy (%)			
Donor	Acceptor	c1mutKSN	c1mutKSS	c2mutKAG	c2mutKAS
HIS57–ND1	ASP81–CG	-	10.34	-	48.99
HIS57–ND1	ASP81–OD1	-	6.38	-	27.41
HIS57–ND1	ASP81–OD2	-	7.64	-	20.04
SER139–OG	LYS136–O	13.14	26.55	6.58	20.44

HIS, histidine; ASP, aspartic acid; SER, serine; LYS, lysine; K, lysine; N, asparagine; G, glycine; S, serine; c1mutKSN = clade I, K80+S91+N174; c1mutKSS = clade I, K80+S91+S174; c2mutKAG = clade II, K80+A91+G174; c2mutKAS = clade II, K80+A91+S174; dash, no significant H-bond occupancies observed.

**Table 3 viruses-11-00691-t003:** Distance between atoms.

Atoms		Distance (Å)			
Donor	Acceptor	c1mutKSN	c1mutKSS	c2mutKAG	c2mutKAS
HIS57–ND1	ASP81–CG	4.97 ± 1.00	5.69 ± 1.75	5.41 ± 1.34	4.95 ± 1.89
HIS57–ND1	ASP81–OD1	5.13 ± 0.96	5.87 ± 1.89	5.76 ± 1.36	4.87 ± 2.14
HIS57–ND1	ASP81–OD2	5.27 ± 1.00	5.83 ± 1.94	5.73 ± 1.37	5.02 ± 2.10
SER139–OG	LYS136–O	4.35 ± 1.81	4.33 ± 1.97	5.20 ± 1.99	4.88 ± 2.12

HIS, histidine; ASP, aspartic acid; SER, serine; LYS, lysine; K, lysine; N, asparagine; G, glycine; S, serine; c1mutKSN = clade I, K80+S91+N174; c1mutKSS = clade I, K80+S91+S174; c2mutKAG = clade II, K80+A91+G174; c2mutKAS = clade II, K80+A91+S174.

**Table 4 viruses-11-00691-t004:** Square inner product of betweenness centrality.

	c1mutKSN	c1mutKSS	c2mutKAG	c2mutKAS
c1mutKSN	1	0.62	0.37	0.42
c1mutKSS		1	0.57	0.47
c2mutKAG			1	0.46
c2mutKAS				1

K, lysine; N, asparagine; G, glycine; S, serine; c1mutKSN = clade I, K80+S91+N174; c1mutKSS = clade I, K80+S91+S174; c2mutKAG = clade II, K80+A91+G174; c2mutKAS = clade II, K80+A91+S174.

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
