# Peer review of "Effects of the Q80K Polymorphism on the Physicochemical Properties of Hepatitis C Virus Subtype 1a NS3 Protease"

_viruses, 2019, doi:10.3390/v11080691_

Round 1

Reviewer 1 Report

The authors have in part responded to my comments. However, there are still some points that can be improved.

My previous comments related to correlation of the observed mutations. I did not mean to ask, whether individual quasispecies in a particular isolate contain these mutations (in positions 91 and 174) together with Q80K. My questions was: do consensus sequences of different isolates contain a pair of mutations in positions 91 *and* 174 simultaneously more frequently than random, as frequently as random, or less frequently as random. This can be computed by comparing the frequency of a double mutation 91+174 with the product of the frequencies of single mutations at positions 91 and 174

The same applies to my question about the phylogenetic distribution of the mutations. *Whithin* the subclade 1A, do the mutations in positions 91 and 174 happen in one branch or in several branches in some sort of correlation with Q80K. For this, the authors will need to build a phylogenetic tree and look at the distribution of the mutations in the leaves. Perhaps also reconstruct ancestral states. I can recommend RAxML to do this, but I leave the choice of the tool to the authors. It is critical to perform this analysis, since if the additional mutations happen only once, we cannot draw any conclusions about their functional importance, and then the whole study setup becomes irrelevant.

It also not exaclty clear to me whether the observed flexibility of the NS4A peptide (discussed in the network analysis part) is consistent between the replicas. Please specify.

Minor comments.

Now I can see from the figure that H57 and S139 are structural neighbors of the mutated residues, but please make it clear in the text

Since considered mutated positions include position 91, I find the notation for the mutants unfortunate. Namely, I suggest renaming c1mutKN to c1mutKSN, c2mutKG to c2mutKAG, etc.

I don't understand the sentence at lines 374-375 "The occurrence of Q80K variants in HCV-1a clade II sequences can be determined by second site polymorphisms, 91 and 174."

The statement "While simeprevir treatment selected resistant mutations such as D168A/V in the wild-type virus, engineered Q80K/R recombinants were responsible for selecting resistant R155K variants." should be referenced.

I find talking about a 0.63-fold inclrease (line 390) confusing. It is a decrease, isn't it?

Line 394: "position-80-substitutions" -> "substituttions at the position 80"

Author Response

June 3, 2019

Mrs. Gloria Gao,

Viruses

Dear Editor:

I wish to submit a revised version of our manuscript (ID 484070) titled “Effects of the Q80K polymorphism on the physicochemical properties of hepatitis C virus subtype 1a NS3 protease” for publication in Viruses.

We would like to thank the Reviewers for their comments, which we have used to improve the manuscript. The new version of our manuscript has been revised in terms of English and its content. Please find enclosed below a response letter with point-by-point answers to the Reviewers’ comments. We hope that the Reviewers find we have addressed all of the issues in a satisfactory way and the manuscript can now be deemed suitable for publication in your journal. We have made the recommended changes/additions as requested. The modifications are highlighted (in red) in the manuscript.

Thank you for your consideration. I look forward to hearing from you.

Sincerely,

Allan Peres-da-Silva

Laboratório de Hepatites Virais, Instituto Oswaldo Cruz/FIOCRUZ

Rio de Janeiro, RJ, Brazil

Tel.: 55-21-2562-1704

[email protected]

 Suggestions for improving the manuscript according to Reviewer 1:

1) My previous comments related to correlation of the observed mutations. I did not mean to ask, whether individual quasispecies in a particular isolate contain these mutations (in positions 91 and 174) together with Q80K. My questions was: do consensus sequences of different isolates contain a pair of mutations in positions 91 *and* 174 simultaneously more frequently than random, as frequently as random, or less frequently as random. This can be computed by comparing the frequency of a double mutation 91+174 with the product of the frequencies of single mutations at positions 91 and 174

            We compared the frequencies of K80+S91+N/S174 mutant sequences with the frequencies of Q80+S91+N/S174 sequences. We expressed the frequencies at which these polymorphisms were found in the sequences. Statistical data were significant.

            We included the following sentences in the manuscript:

-          Material and Methods (lines 78-89):

Based on the HCV-1a clade I and II amino acid signature pattern, only clade I sequences presented high frequencies of K80, S91, and N/S174. To verify if K80+S91+N/S174 polymorphisms were more frequent in the same clade I sequences than based on a random distribution, we estimated the frequencies of K80+S91+N174 and K80+S91+S174 and then compared the results with the frequency of clade I Q80+S91+N174 and Q80+S91+S174 polymorphisms. To this end, we considered 116 sequences, which contained either K or Q in position 80; the remaining four sequences had either G, L, N or R at this site. A chi-square test was performed to verify the significance of sample size, with p<0.05 being considered statistically significant. The total sample size of Q80 (44/116, 37.9%) and K80 (72/116, 62.1%) polymorphisms and their frequencies was estimated. A chi-square test was used to compare the observed counts of each category (K80+S91+N174, Q80+S91+N174 and K80+S91+S174, Q80+S91+S174) with their respective expected counts, specified by probability distributions.

-          Results (lines 179-192):

3.1. Sequence analysis, statistical treatment, and comparative modeling

Different amino acid signature frequencies were documented at positions 80, 91, and 174 of NS3 protease across HCV-1a clades. Q80 (44/120, 36.7%), K80 (72/120, 60.0%), S91 (43/120, 35.8%), N174 (97/120, 80.8%), and S174 (22/120, 18.3%) were found at high frequencies in HCV-1a clade I. In clade II, the most prevalent residues at these sites were: Q80 (117/120, 97.5%), A91 (116/120, 96.7%), S174 (86/120, 71.7%) and G174 (15/120, 12.5%). These polymorphic frequencies are in accordance with a recent analysis of more than 2000 HCV-1a NS3 sequences from both clades [13].

We observed 30 isolates containing S91+N174, of which 12 (40.0%) were K80+S91+N174 and 18 (60.0%) were Q80+S91+N174 (p=0.0127). Another 13 isolates contained S91+S174, of which, 12 (92.3%) were K80+S91+S174 and one (7.7%) was Q80+S91+S174 (p=0.0246). It should be noted that there were also sequences containing K80 (n=48), Q80 (n=25), G80 (n=1), L80 (n=1), N80 (n=1), and R80 (n=1) residues. These sequences presented a broad polymorphic variety in their NS3 sequence, including at sites 91 and 174, and were not incorporated in the structural analysis of the present study.

2) The same applies to my question about the phylogenetic distribution of the mutations. *Whithin* the subclade 1A, do the mutations in positions 91 and 174 happen in one branch or in several branches in some sort of correlation with Q80K. For this, the authors will need to build a phylogenetic tree and look at the distribution of the mutations in the leaves. Perhaps also reconstruct ancestral states. I can recommend RAxML to do this, but I leave the choice of the tool to the authors. It is critical to perform this analysis, since if the additional mutations happen only once, we cannot draw any conclusions about their functional importance, and then the whole study setup becomes irrelevant.

            The previous results are now integrated with phylogenetic analysis, which shows that Q80K polymorphism and S91+N/S174 substitutions arose in viral lineages of two independent subclades (IA and IB), rather than branching out from a unique ancestral node. For the this, we chose to use the PhyML 3.0 program.

            We included the following sentences in the manuscript:

-          Material and Methods (lines 169-176):

2.6. Phylogenetic analysis

To investigate possible correlations between clade I S91+N/S174 polymorphisms and the K80 phenotype, phylogenetic analysis was performed with the same HCV-1a clade I and II reference sequences used to determine specific amino acid signature patterns. Maximum-likelihood phylogenetic trees were inferred using the PhyML program [30] with the approximate likelihood-ratio test (aLRT) [31] based on a Shimodaira-Hasegawa-like procedure. Specifically, general time reversible was used as substitution model and a subtree pruning and regrafting-based tree search algorithm was used to estimate tree topologies.

-          Results (lines 354-361):

3.4. Phylogenetic analysis

As shown in Figure 8, the phylogenetic tree was divided into two major clades, with K80+S91+N/S174 sequences clustered exclusively in clade I within two distinct monophyletic subclades, IA and IB. Subclade IA included 16 K80 isolates, half of which contained S91+N174 residues and half S91+S174 residues. This subclade contained also 16 wild-type sequences (Q80+S91+N/S174), which were grouped with high statistical support (aLRT=0.90). In contrast, subclade IB comprised 27 HCV-1a sequences, of which four presented the polymorphic sequence K80+ S91+N174 and another four the K80+S91+S174 sequence.

-          The new figure 8 was added with the following caption (lines 363-371):

Figure 8. ML phylogenetic tree of HCV-1a based on 120 representative sequences of each, clade I (light yellow) and clade II (gray). Within clade I, subclade IA and subclade IB are highlighted by a solid black line. A total of 12 K80+S91+N174 (red) and 12 K80+S91+S174 (blue) polymorphic sequences are grouped in the two subclades, whereas most of the 19 wild-type sequences (brown and green) cluster into a single branch, formed by 16 sequences. In clade I, branches colored in black are represented by sequences containing K80 (n=48), Q80 (n=25), G80 (n=1), L80 (n=1), N80 (n=1), and R80 (n=1). In clade II, black branches represent sequences containing Q80 (n=117), R80 (n=2), and L80 (n=1). The tree was rooted with HCV-1b strain (D90208). Approximate likelihood-ratio test values ≥0.90 are shown. For better visualization, sequence names were removed.

-          Discussion (lines 383-385):

In our analysis, Q80K polymorphism correlated with S91+N/S174 substitutions. Importantly, these residues arose in viral lineages of two independent subclades (IA and IB), rather than branching out from a unique ancestral node…

3) It also not exactly clear to me whether the observed flexibility of the NS4A peptide (discussed in the network analysis part) is consistent between the replicas. Please specify.

            We agreed with the reviewer and added the following sentence in the manuscript (lines 309- 310): These results were consistent among replicas, indicating that the observed differences were real and reproducible

Minor comments.

4) Now I can see from the figure that H57 and S139 are structural neighbors of the mutated residues, but please make it clear in the text

We included the following sentence in the discussion section (lines 398-399): Importantly, residues H57 and S139 are structural neighbors of polymorphic sites 91 and 174…

5) Since considered mutated positions include position 91, I find the notation for the mutants unfortunate. Namely, I suggest renaming c1mutKN to c1mutKSN, c2mutKG to c2mutKAG, etc.

            We modified the notations for the structures throughout the manuscript, tables and figures.

6) I don't understand the sentence at lines 374-375 "The occurrence of Q80K variants in HCV-1a clade II sequences can be determined by second site polymorphisms, 91 and 174."

            For better clarity, we have removed this sentence from the manuscript. The new paragraph is (lines 425-428):

The NS3 proteolytic activity and consequently viral replication could be compromised by modifying the physicochemical properties of the NS3 protease in the presence of the Q80K mutation, potentially hindering its emergence in the clade II viral quasispecies population and explaining why this mutation is not detected in these sequences.

7) The statement "While simeprevir treatment selected resistant mutations such as D168A/V in the wild-type virus, engineered Q80K/R recombinants were responsible for selecting resistant R155K variants." should be referenced.

            We insert the reference [36]. (line 438)

8) I find talking about a 0.63-fold inclrease (line 390) confusing. It is a decrease, isn't it?

            We rephrase the sentence to: The replication capacity of Q80K variants was 1.26-fold higher than that of the wild-type virus. However, after the emergence of R155K variants (Q80K+R155K), the replication capacity was only 0.63-fold higher than that of the wild-type virus [36]. (Lines 440-442).

9) Line 394: "position-80-substitutions" -> "substituttions at the position 80"

            We made the modification as suggested. (lines 444-445)

Reviewer 2 Report

The authors have sufficiently addressed comments in writing. Additional laboratory experiments to confirm the results obtained by modeling were not done, since considered outside the scope of this study, but relevant literature was cited and discussed.

The following minor issues remain to be addressed:

- given that new content has been added, the two paragraphs (line 157-163 versus line 164-168) do not connect anymore. This could be improved by swapping order of sentences in the 2nd paragraph (swap line 164-166 with line 166-168).

- Line 216 should read "fluctuation values"

- Line 385-401: The language needs more work. Referring to HCV genotype 1a as "HCV1-1a" in this paragraph is suboptimal, since one isolate is called "HCV1". Thus, I would use "HCV genotype 1a" and then the specific isolate names. The Genbank number of the cell culture infectious TN clone used in this study is JX993348.1. In line 396, it would be good to state that this isolate  originally had K at position 80.

-Line 404/405: should read "variants resistant to DAA inhibitors".

-Line 417: Replace "variants" by "substitutions"

-Line 419: do you mean "differences in frequencies in DAA resistance substitutions"?

In general, English language can still be improved

Author Response

June 3, 2019

Mrs. Gloria Gao,

Viruses

Dear Editor:

I wish to submit a revised version of our manuscript (ID 484070) titled “Effects of the Q80K polymorphism on the physicochemical properties of hepatitis C virus subtype 1a NS3 protease” for publication in Viruses.

We would like to thank the Reviewers for their comments, which we have used to improve the manuscript. The new version of our manuscript has been revised in terms of English and its content. Please find enclosed below a response letter with point-by-point answers to the Reviewers’ comments. We hope that the Reviewers find we have addressed all of the issues in a satisfactory way and the manuscript can now be deemed suitable for publication in your journal. We have made the recommended changes/additions as requested. The modifications are highlighted (in red) in the manuscript.

Thank you for your consideration. I look forward to hearing from you.

Sincerely,

Allan Peres-da-Silva

Laboratório de Hepatites Virais, Instituto Oswaldo Cruz/FIOCRUZ

Rio de Janeiro, RJ, Brazil

Tel.: 55-21-2562-1704

[email protected]

Suggestions for improving the manuscript according to Reviewer 2:

1) Given that new content has been added, the two paragraphs (line 157-163 versus line 164-168) do not connect anymore. This could be improved by swapping order of sentences in the 2nd paragraph (swap line 164-166 with line 166-168).

            We swapped the order of the sentences as suggested. (Lines 193-197)

2) Line 216 should read "fluctuation values"

            The new sentence is (line 245): These residues are located close to the NS4A cofactor, which is probably the reason for the elevated fluctuation values…

3) Line 385-401: The language needs more work. Referring to HCV genotype 1a as "HCV1-1a" in this paragraph is suboptimal, since one isolate is called "HCV1". Thus, I would use "HCV genotype 1a" and then the specific isolate names. The Genbank number of the cell culture infectious TN clone used in this study is JX993348.1. In line 396, it would be good to state that this isolate originally had K at position 80.

            We are now referring "HCV genotype 1a" in this paragraph (lines 435-452). Also, we changed the Genbank number of TN clone to JX993348.1 (Line 448) and stated that this isolate originally had K at position 80: Interestingly, for isolate TN, which originally harbored K at position 80, viral spreading kinetics… (lines 446, 447)

4) Line 404/405: should read "variants resistant to DAA inhibitors".

            We made the modification as suggested. (lines 455-456)

5) Line 417: Replace "variants" by "substitutions"

            We also made the modification as suggested. (line 468)

6) Line 419: do you mean "differences in frequencies in DAA resistance substitutions"?

            The new sentence is (lines 469-470): It is not yet fully understood whether the differences in frequencies in DAA resistance substitutions found…

7) In general, English language can still be improved

We sent the manuscript for English review with editage.com company (https://www.editage.com/). We are also submitting the certificate of revision.

Reviewer 3 Report

The authors appear to have adequately addressed the reviewer comments in their revised manuscript.

Author Response

no further action required

Round 2

Reviewer 1 Report

The authors have adquately addressed all my comments, which, I hope, could improve the study. I would perhaps use Fisher's exact test, as it has less parameters than chi-squared, and consider the two groups Q80+S91+N174 and Q80+S91+S174 together in the statistical analysis, but this is a matter of personal taste, and does not diminish the quality of the paper.

This manuscript is a resubmission of an earlier submission. The following is a list of the peer review reports and author responses from that submission.

Round 1

Reviewer 1 Report

In this manuscript, Peres-da-Silva and colleagues evaluate HCV subtype 1a NS3 polymorphisms by molecular dynamics and network analyses. They focus on the Q80K variant found predominantly in clade I of NS3 in HCV-1a, which has been associated with reduced simeprevir inhibition. In particular, they evaluate the impact of Q80K on other naturally occurring NS3 polymorphisms (S/A91 and S/N/G174) to predict effects on NS3 structure and function. The MD simulations revealed that in the case of the K80 variant, the polymorphisms led to variations in hydrogen bond occupancies and flexibility of active site residues. Therefore, the polymorphisms could affect NS3 protease activity. The article addresses an interesting question as to how natural polymorphisms within HCV non-structural proteins affects DAA activity and the emergence of DAA resistant-variants. However, the article is quite technical and perhaps more suited for a specialist journal than for a general virology audience. Inclusion of further experiments would make it more suited for the readership of the journal:

Specific comments:

1.     It would be interesting to evaluate the proteolytic activity of the NS3 mutants (e.g. in vitro assay with recombinant NS3) to confirm whether the differences in the flexibility of the H57 catalytic residue indeed impair enzymatic activity. Furthermore do the polymorphisms affect the activity of protease inhibitors in such assays?

2.     Do the polymorphisms evaluated in this manuscript affect replication fitness of the virus (e.g. in a replicon model)?

Minor comments:

1.     It was unclear from the text what exactly the model names were describing (e.g. c1mutKN, c1mutKS, c2mutKG, c2mutKS) – would recommend making this readily apparent in the text rather than requiring the reader to figure it out from the figures/ figure legends

2.     Figure 1 – the black background in the alignment makes the features being pointed out less clear. I would recommend white background.

Reviewer 2 Report

Naturally occurring polymorphisms at NS3 protease position 80 have been shown to influence efficacy of treatment with HCV protease inhibitors in vivo and in vitro and can thus act as resistance associated substitutions (RASs).

Big regional differences in prevalence of position 80 polymorphisms have been described. In the enclosed study, the authors provide evidence that occurrence of Q80K polymorphisms might depend on occurrence of second site polymorphisms in the NS3 protease. The authors use various modeling approaches to confirm the influence of specific second site polymorphisms on the effect of the Q80K polymorphism on structure and function of the NS3 protease.

The study is clinically relevant and novel since it describes a link between occurrence of a RASs and second site polymorphisms.

A major drawback of this study is that findings in this study have not been confirmed by in vitro assays, e.g. enzymatic assays, however, this might be outside the scope of this study.

Minor specific issues in Introduction:

Line 36-37: Improve formulation. “….drugs directly targeting HCV proteins, so called direct-acting antivirals (DAAs)”

Line 39: Is reference 3 here appropriate? It mainly relates to the PI simeprevir, which is not frequently used in clinical practice anymore. Other, more recent reviews giving an overview of currently used protease inhibitors and associated antiviral resistance might be more appropriate (e.g. Sarrazin, J Hepatol, 2016, which is also covering NS3 protease position 80 polymorphisms; Pawlotsky, Gastroenterology, 2016; Sorbo, Drug Resist Updat, 2018).

Line 44-45: this statement could be broadened: “…and has been associated with reduced efficacy of simeprevir but also other protease inhibitors in vivo and in vitro”. In vivo literature is reviewed in Sarrazin, J Hepatol, 2016 and Pham, J Hepatol, Epub ahead of print. In vitro data are reported by Pham, J Hepatol, Epub ahead of print.

Line 46-47: The authors should acknowledge that Q80K is mediating a low level of PI resistance and mainly to simeprevir. However, it might facilitate escape from other PIs by facilitating selection of additional high level resistance RASs. E.g “….its presence could facilitate emergence of RASs conferring higher levels of resistance to protease inhibitors or other DAAs” (Evidence for the first part of the statement, that pre-existing Q80K results in accelerated escape from PIs by co-selection of high-level resistance RASs is reported in Pham, J Hepatol, Epub ahead of print)”.

Line 51: Is a bit unclear. Do you mean “…possibility of a certain HCV-1a clade” / “…possibility of one of these HCV-1a clades”?

Minor specific issues in Results:

The descriptions in the Results section are very technical. It is possible to follow by reading Materials and Methods and Discussion in parallel. However, for those who are not experts in these kind of modelling approaches, it would be nice with better explanations of what the technical findings indicate. E.g. Line 182, 185, 188 what will the RMSF tell us, what does higher fluctuation and more flexibility indicate? Line 203, what does high occupancy indicate.

Line 152-154: It might be nice to explain briefly how these sequences were obtained, possibly a short version of lines 76-88 in Materials and Methods. Especially, it would be good to clarify here that the K at position 80 was not found in these genbank sequences but “mutated” prior to analysis – this could possibly be also indicated by a different colour (red is also used for position 174, which is reported as found in genbank).

Line 160-161: This is a bit unclear: Which crystal structure? The one of the selected reference? Should it be “clade I sequences” and “clade II sequences”, since thre are 2? In Figure 1, which is cited, there is no crystal structures shown.

Line 170: “c1mutNS” – is this one not included in Figure 1?

Minor specific issues in Discussion:

Line 317-322: The first statement relating to differential barrier to resistance for gt 1a versus 1b is a bit out of context. The described difference is mostly true for treatment regimens including “old PIs” such as telaprevir and boceprevir, as also discussed in the cited reference. It is caused by acquisition of RAS R155K requiring 2 nucleotide changes for gt 1b versus 1 nucleotide change for gt 1a. Thus, the mechanism underlying this differential barrier of resistance is quite different from the possible mechanisms described in this study.

Instead, the authors could focus on emphasizing that they find that second site polymorphisms determined the occurrence of the Q80K RASs. To my knowledge this has not been described before.

However, in this context they might want to cite Sun, Hepatology, 2012. In this study, the authors find that second site polymorphisms have an impact on the emergence of resistance to an NS5A inhibitor by increasing resistance conferred by an otherwise low level resistance RASs.

Line 324: The authors are missing to mention infectious cell culture systems (in addition to subgenomic replicons), as used in the recent study by Pham, J Hepatol, Epub ahead of print. It might be worthwhile looking into aa residues at NS3 protease position 91 and 174 in published genotype 1a replicons and infectious HCV recombinants with engineered Q80K and correlate findings with the reported impact on fitness.

Line 328-335: This section is a bit unclear, especially lines 332-335.

Reviewer 3 Report

The manuscript by Peres-da-Silva and co-authors presents an analysis of molecular dynamics simulations of several potentially clinically relevant mutants of the NS3/4A complex from the Hepatitis C virus. This is definitely an important topic, and the approach of the authors, which combines sequence analysis to select the mutants with structural simulations is relevant to the problem and novel in the field. The manuscript is generally well written and easy to follow. However, I have several significant concerns listed below that have to be addressed before the manuscript can be considered for publication.

Major points.
1. First, and most importantly, I could not take it out from the text, how many replicas did the authors perform? It is customary in these studies to perform 4 to 8 replicas starting from the the same structure and running the simulation independently. This allows to assess the significance of the observed differences between the systems. 100 ns is not a super long simulation, but it would be all right, if the replicas were made.
2. The other concern pertains to the sequence analysis part of the study. First of all, I don’t understand why the authors restrained themselves to only considering the sequences from Pickett et al., which were collected in 2008, i.e. more than 10 years ago. Why not consider all sequences from a database, e.g. LANL HCV database or HCVDB? Genotyping tools work well enough, so it will be no problem assigning genotypes and sub-genotypes to them. Second, correlations of the observed mutations should checked. That is, does S91 occur in the same sequences at N/S174, and are they correlated with Q80K? Can these substitutions be explained by viral phylogeny, i.e. do they happen in one branch of the phylogenetic of NS3 sequences or independently in several branches?
3. The choice of the structure to run simulations. There are structures with better resolution, but typically in complex with a ligand. Is that an issue for simulations? Why was 2f9v chosen, despite its relatively poor resolution?
4. From Table 1, it appears to me that only mutations at position 174 were considered. What about position 90 that is discussed at length in the manuscript? Or do I get it wrong, and simulations with mutations at site 90 were also performed? In that case this should be clarified.
5. Network analysis reveals mainly differences in binding of the NS4A peptide. However, I an not convinced that this interaction was modelled reliably, and I doubt that any speculations here are useful. The simulation is too short, and the NS4A peptide is not stable, as evident from Fig 3b. It might be however the case that the observed differences are real and reproducible, but we need more replicas to prove that. Anyway, the relation between the mutated sites that lie in a spatial distant region from the peptide and the peptide-protein interaction is not explained in the manuscript.

Minor points.
1. p. 1, l. 33: HCV is rather a global health threat, not an issue. This statement could also use a reference.
2. p. 2, l. 48-52: this passage looks somewhat out of place here.
3. p. 6, l. 183: the residues 80, 91, and 174 do not fluctuate much, but what about the neighbouring residues? I could imagine that they need to move to accommodate the mutations. Same paragraph, residues 14 and 15 that do fluctuate, they are not neighbours of 80, 91, and 174, are they? So what would be an explanation for their behaviour?
4. Table 2: what does a dash represent?
Figure 5: NS4A peptide should be coloured differently in the 3D structures.
5. p. 10, l. 264: Can the authors comment on the observed CIP values? What does it entail for the similarity of the trajectories?
6. p. 11, l. 277-280: repetitive passage, cf. previous
7. p. 12, l. 318-320: this is not the definition of genetic barrier